# Copernicus in Support of Monitoring, Protection, and Management of Cultural and Natural Heritage

**Alessandra Bonazza** [1] , **Nico Bonora** [2,*] , **Benjamin Duke** [3], **Daniele Spizzichino** [2], **Antonella Pasqua Recchia** [4] and **Andrea Taramelli** [2,5]

1   National Research Council of Italy, Institute of Atmosphere Sciences and Climate, 40126 Bologna, Italy; a.bonazza@isac.cnr.it
2   Italian Institute for Environmental Protection and Research, 00176 Rome, Italy; daniele.spizzichino@isprambiente.it (D.S.); andrea.taramelli@isprambiente.it (A.T.)
3   Deutsches Archäologisches Institut, 14195 Berlin, Germany; benjamin.ducke@dainst.de
4   Italian Ministry of Culture, 00186 Rome, Italy; recchia@beniculturali.it
5   Istituto Universitario di Studi Superiori di Pavia (IUSS), University School for Advanced Studies Pavia, 27100 Pavia, Italy
*   Correspondence: nico.bonora@isprambiente.it

**Abstract:** The current Copernicus evolution aims to meet horizontal users' needs in order to widen uptake of the Copernicus monitoring products by non-traditional users. In 2019, the European Commission initiated a coordinated action to evaluate the current and potential uptakes of Copernicus products, and for the monitoring and protection of European Cultural and Natural Heritage in a future climate change scenario. An interaction matrix was developed, circulated to and fulfilled by users in order to collect their needs and identify the main gaps in terms of monitoring data and information. The results show what users require from Copernicus to face the daily challenges of preserving and protecting CH features. Moreover, the interaction with users identified a data and information access model that best maximizes uptake by the users. The present work illustrates the user requirement coordination mechanism adopted by the European Copernicus Cultural Heritage Task Force; synthesises the results achieved in terms of gap analysis; and assesses the current and potential uptake of Copernicus data, services, and products in support of the monitoring and protection of European cultural heritage. It also provides recommendation about the implementation of infrastructural solutions to improve Copernicus services data and information access by cultural heritage users.

**Keywords:** earth observation; spatial and temporal resolution; safeguarding heritage; climate change; conservation

## 1. Introduction

During recent decades, awareness of the need for efficient, science-based tools to monitor and protect cultural and natural heritage has rapidly grown. Indeed, heritage assets are increasingly at risk because of the impact of natural and anthropogenic hazards, the frequency and intensity of which continue to be amplified by climate change [1–4]. The protection of archaeological sites and monumental complexes in the age of mass tourism and climate change represents a growing challenge, which can only be addressed by integrating management models and practices. In this context, the innovative application of remote sensing technologies [5] and Copernicus data and information could certainly constitute a turning point, as demonstrated in other transversal areas [6]. Sites and monuments are affected by various environmental agents, acting in synergy, which leads to varying frequency and intensity. The majority of these agents, such as wind erosion [7–9], ground water level changes [10], air pollution, and climate change [7,9], can be extremely harmful when they affect a site over a long period of time [11]. Therefore, long-term monitoring

of prioritised environmental and climate parameters and indicators, at proper spatial and temporal resolution, is a key requirement for setting up action plans and strategies for sustainable management [12]. This monitoring should rely on the integration of data from remote sensing and in situ measurements, along with climate modelling outputs. Such an integrated approach is a prerequisite for decreasing the vulnerability of cultural heritage in all steps of the risk management cycle: prevention/preparedness, emergency and recovery [13–15]. Among the many sources of data available, Europe has been delivering a series of free and open satellite-derived data and modelling information through the Copernicus program, linking the implementation of different policies of the Union to the use of such resources [16–18]. Despite their value, the adoption of Copernicus EO data and information, especially from non-technical local and regional governmental authorities, remains low due to a general asymmetry of information between offer and demand, especially among public administrations [19]. Thus, within the Copernicus infrastructure, the possibility of tailoring and clustering services, data, and products, in order to satisfy the cross-cutting requirements of the cultural heritage community, constitutes an historic opportunity for development and an evolution towards their uptake by horizontal/non-traditional users [19–21]. For this purpose, an initial EC study (Copernicus services in support of cultural heritage) was commissioned by the European Commission's Directorate General for Internal Market, Industry, Entrepreneurship, and SMEs (DG GROW) [22]. This was the initial step in starting an institutional action to investigate the possibility of including specific services to satisfy the needs of cultural heritage preservation, monitoring, and management in the Copernicus portfolio. In 2019, a task force was formalised by the Copernicus Committee to evaluate the uptake extent in the field of cultural heritage management. The aims of the Copernicus Cultural Heritage Task Force (CCHTF) were to consolidate the outcomes of the aforementioned study; to assess the current and future potential of Copernicus data, services, and products uptake by users; and to identify possible Copernicus architectural solutions to support data and/or information access. The CCHTF was composed by the member states' (MS) national experts, from both the cultural heritage and Earth observation domains, officially coordinated by Italy and chaired by the Italian Ministry of Culture—MiC (formerly the Italian Ministry of Cultural Heritage and Activities and Tourism—MiBACT) [23]. The CCHTF included an extended range of stakeholders (research and business communities, public authorities, policy and decision makers, operational bodies, and social players) who provided a set of user needs, extending across the different cultural heritage disciplines. This paper illustrates the methodological approach adopted and the results obtained by the activities of the CCHTF, in order to achieve the following objectives:

(i)   Map the user requirements, as provided by the MS delegates, for cultural heritage in the Earth observation domain;

(ii)  Analyse how existing Copernicus data, services, and products could satisfy those requirements;

(iii) Identify possible enhancement and customisation options within already operational Copernicus Core Services.

## 2. Materials and Methods

The overall concept underlying our work is illustrated in Figure 1, where the methods, tools, and approaches applied for the user needs analysis, matched with the requirements with Copernicus capacities, and a gap analysis, are described.

**Figure 1.** Flow chart of the methodological approach.

## 2.1. User Needs Analysis and Requirements Definition

In order to achieve a comprehensive analysis of the user needs and to derive the relevant technical requirements, we capitalised on the results of the consultation process behind the PricewaterhouseCoopers (PwC) study [22]. The study involved an extended range of stakeholders in the consultation process and provided a set of user needs scattered among different cultural heritage disciplines, not specifically centered on the application of Earth observation services and data. As the first step in the CCHTF methodology, a specific survey addressed to 48 stakeholders, represented by the CCHTF members from different target groups (e.g., scientific communities, institutions, public authorities, and civil protection—Table 1), was conducted with the aim of collecting updated user needs and identifying un-scattered measurable requirements (Steps 1 and 2, Figure 1).

**Table 1.** List of consulted national users.

| Country | | Consulted Institution/Authority/Entity |
|---|---|---|
| BE | - | Service public de Wallonie—DGo4—Agence Wallonne du Patrimoine—Direction de l'Appui Scientifique et Technique |
| CY | - | Cyprus University of Technology |
| | - | Department of Electromechanical Services |
| CZ | - | Department of Anthropology, University of West Bohemia in Pilsen |
| | - | Department of Applied Geoinformatics, Czech University of Life Sciences Prague |
| DE | - | Deutsches Archäologisches Institut (DAI) |
| | - | Stiftung Preußischer Kulturbesitz (SPK) |
| | - | Auswärtiges Amt (AA) |
| | - | Beauftragte für Kultur und Medien (BKM) |
| | - | Bundesministerium für Verkehr und Information (BMVI) |
| | - | Geoforschungszentrum Potsdam (GFZ) |
| | - | Deutscher Verband für Archäologie (DVA) |
| ES | - | Centro para el Desarrollo Tecnológico Industrial (CDTI) |
| | - | Ministerio de Cultura y Deporte |

**Table 1.** *Cont.*

| Country | Consulted Institution/Authority/Entity |
|---|---|
| FR | - Ministère de la Culture (MC) |
| | - Laboratoire de Recherche des Monuments Historiques (LRMH) |
| | - Centre de Recherche et de Restauration des Musées de France (C2RMF) |
| | - Centre national de préhistoire (CNP) |
| GR | - Ministry of Digital Governance |
| IT | - Ministry of Cultural Heritage and Tourism |
| | - ISPRA—Italian Institute for Environmental Protection and Research |
| | - CNR ISAC—National Research Council of Italy Institute of Atmospheric Sciences and Climate |
| | - ISCR -Istituto Superiore per la Conservazione e il Restauro |
| | - ASI—Italian Space Agency |
| | - National Archaeological Parks (Pompei, Colosseum, Ostia Antica) |
| MT | - Department of Civil Protection |
| | - Superintendence of Cultural Heritage |
| | - University of Malta |
| NL | - Rijksdienst voor het Cultureel Erfgoed (RCE) |
| | - Staatsbosbeheer (SBB) |
| | - Convent van gemeente-archeologen |
| | - RAAP (Commercial Archeological Company) |
| NO | - Klima- og miljødepartementet (KLD) |
| | - Riksantikvaren (Directorate for Cultural Heritage) |
| | - Norwegian Institute of Bioeconomy Research (NIBIO) |
| | - Norwegian University of Science and Technology |
| | - The Norwegian Institute for Cultural Heritage Research (NIKU) |
| | - The Norwegian Water Resources and Energy Directorate |
| PL | - Ministerstwo Kultury |
| PT | - FCT Fundação para Ciência e a Tecnologia |
| | - Universidade de Évora |
| | - Direção Regional de Cultura do Alentejo |
| UK | - Historic England |
| | - English Heritage |
| | - The National Trust |
| | - Historic Environment Scotland |
| | - The Royal Commission on the Ancient and Historical Monuments of Wales—RCAHMW |

The survey consisted of the following questions:

1. How important is this information for the daily management of your work?—Priority dimension (relative weight from 0 to 5);
2. How frequently do you need or want to check this parameter?—Temporal dimension;
3. How accurate must this part be for your purposes?—Spatial dimension.

Consequently, the 73 user needs identified were grouped into the nine monitoring domains identified by the analysed PwC study [22] (Table 2).

**Table 2.** Monitoring domains identified.

| n. | Monitoring Domains |
|---|---|
| 1 | Detection of underground archaeological sites through the study of the natural environment |
| 2 | Non-destructive analysis of the underground/underwater positioning of cultural heritage features |
| 3 | Non-destructive analysis of the surface positioning of cultural heritage features |
| 4 | Mapping of the cultural landscape of the site and identification of the specific risks to which it is exposed |
| 5 | Monitoring of the evolution of the natural environment of cultural heritage sites |
| 6 | Monitoring of the evolution of the natural environment of the natural heritage sites |
| 7 | Observation of changes on the built structure of cultural heritage sites |
| 8 | Recommendations for facilitating emergency interventions |
| 9 | Enabling public access to a site |

As the first validation step, the survey was returned by the task force members to the national stakeholders responsible for cultural heritage monitoring and management. The final survey was returned to national stakeholders in the form of a matrix, for their feedback on the correctness of the identified common requirements supportable by Earth observation in terms of geo-spatial services.

*2.2. Validation Activity and Gap Analysis*

In order to understand the extent of the Copernicus potential for user uptake by the cultural heritage community, the user requirements were analysed by the Copernicus entrusted entities, so as to match up the current and planned programme capacities with the identified requirements (Step 3, Figure 1).

All the collected answers from the complete whole matrix were statistically elaborated in order to plot the main clusters and results and to provide recommendations through gap analysis implementation both for routine and on-demand services (Step 4, Figure 1).

**3. Results**

*3.1. User Needs Analysis and Requirements Definition*

The results show that most user needs fall into the "Monitoring of the evolution of both the natural environment of the cultural heritage and natural heritage sites" monitoring domain (Table 2 and Figure 2). The user needs that are less represented and more difficult to cover by remote sensing technology are obviously those relating to underground sites (limited penetration capacity of satellite-borne sensors), and those relating to small structural changes (limits of currently available spatial resolution). The remaining user needs are equally distributed across all other monitoring domains.

Figure 3 provides a view of the link between the monitoring domains and the user requirements already met by Copernicus, derived from expressed needs, with an indication of the service through which they should be covered.

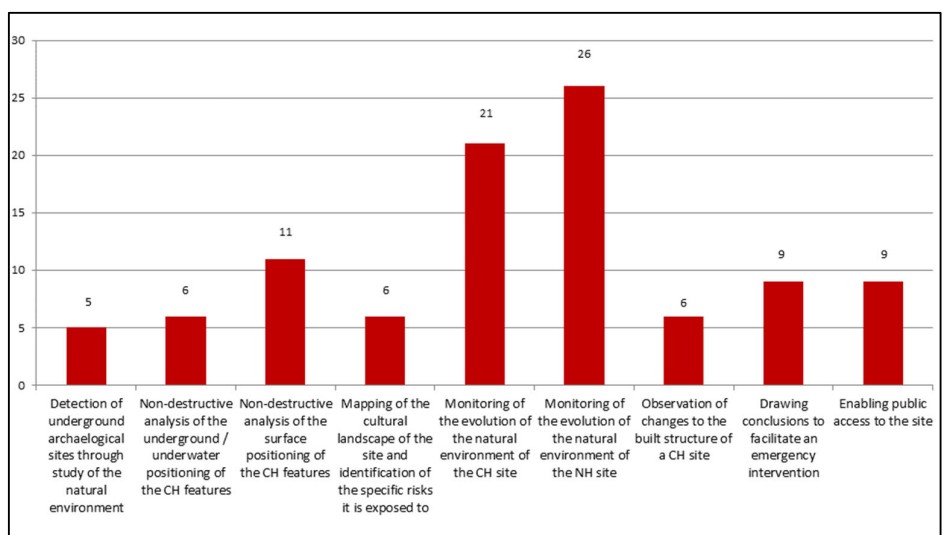

**Figure 2.** The number of user needs associated with each of the nine monitoring domains. The total amount (99 user needs) plotted reflects the presence of the same needs in different domains.

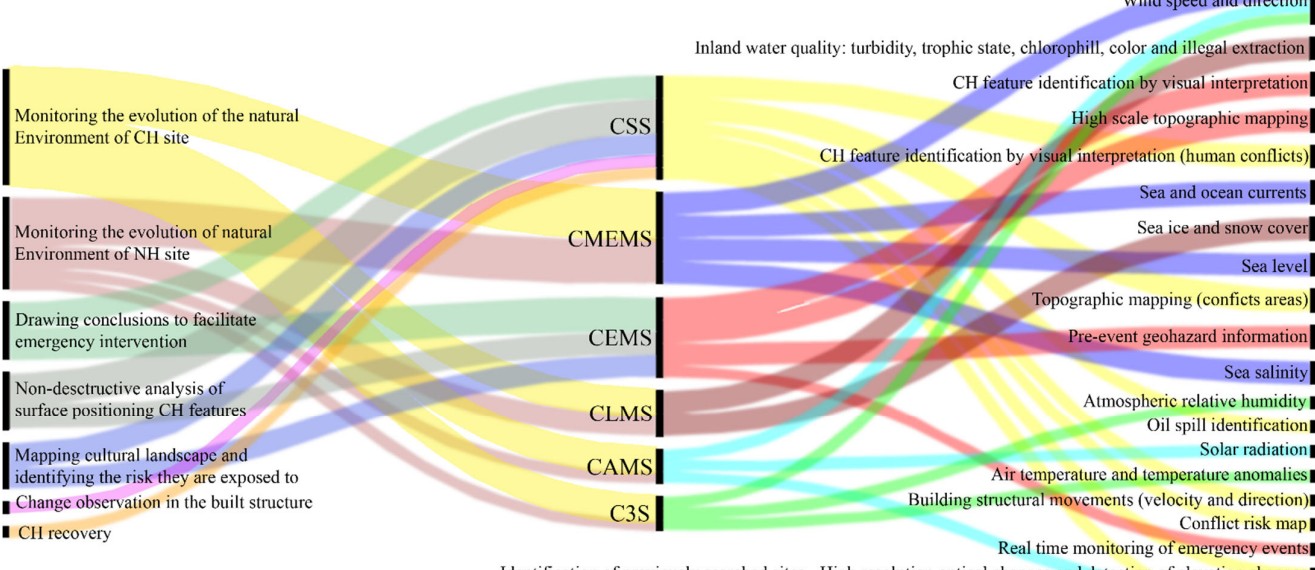

**Figure 3.** Link between high-level user needs (monitoring domains), Copernicus Core services, and user requirements already met by Copernicus. Copernicus Security Service (CSS); Copernicus Service (CEMS); Copernicus Land Monitoring Service (CLMS); Copernicus Atmosphere Monitoring Service (CAMS); Copernicus Climate Change Service (CCCS).

The graphical method chosen to represent the results of the feedback analysis is the Sankey diagram; this is a flow diagram that relates entities through lines whose thickness expresses the quantitative relationship between them. This diagram is particularly suited to highlight and assess the priorities given by user communities to specific requirements.

### 3.2. Validation Activity and Gap Analysis

The interaction Marine Environment Monitoring Service (CMEMS); Copernicus Emergency Management between the Copernicus entrusted entities responsible for the service, space, and in situ component developments also resulted in the identification of a number of further Copernicus products, suitable for support of CH user activities; these mostly

stem from the global land component, atmosphere, climate change, and marine monitoring services, as well as emergency and security (Figure 3 and Tables 3–5).

**Table 3.** Synthesis of entrusted entities consultation, listing the available Copernicus products already suitable to support cultural heritage user requirements, and products that could support cultural heritage users, if improved.

| Service | Requirements Supported by Current Products | Requirements Supportable by Current Products (With Improvements) |
|---|---|---|
| CLMS | | Raster elevation data—elevation change layer |
| | Sea ice and snow cover layers | |
| | Inland water quality information (turbidity, trophic state/chlorophyll, apparent colour, and illegal abstraction) | |
| | | NDVI layer |
| | | Vegetation and vegetation change layers, including infesting vegetation |
| | Forest/tree coverage layer | |
| | | Ground motion layer and data analysis |
| | | Coastal erosion layer—sedimentary balance |
| | | Hydrological changes and network changes layers |
| C3S | | Hydrological changes and network changes layers |
| | Atmospheric relative humidity layer | |
| | Air temperature and temp. anomaly layers | |
| C3S/CAMS | Wind speed and direction layers | |
| CAMS | Pollutant concentration map/model—$NO_2$–$NO$–$SO_2$–$O_3$—PM10–2.5 | 1 to 5 km spatial resolution for built environment |
| | Solar radiation layer | |
| CMEMS | Wind speed and direction layers | |
| | Sea salinity layer | |
| | Sea and ocean current layer | |
| | Sea level layer | |
| CEMS | Pre-event geohazard information | |
| | Real-time monitoring of emergency events | |
| | CH feature identification by visual interpretation | |
| | | Hydrological forecast information |
| | Large-scale topographic mapping | |
| CSS | Topographic mapping | |
| | CH feature identification by visual interpretation (Human conflict risk monitoring could satisfy this requirement) | |
| | Identification of previously searched sites in the area High-resolution elevation change Optical change detection | |
| | Building structural movements, velocity and direction | |
| | Conflict risk map | |
| | Oil spill identification | Pollutant concentrations (hyperspectral capacity required) |
| | Vessel identification (smuggling and recovery actions) | |

**Table 4.** Description of requirements that can be satisfied by improving current Copernicus products.

| Service | Requirements Supportable by Current Products (with Improvements) | Requested Improvement | | | |
|---|---|---|---|---|---|
| | | Spatial Resolution | | Frequency of Update | |
| | | Current | Required | Current | Required |
| CLMS | Raster elevation data—elevation change layer | 25 m resolution with vertical accuracy: +/− 7 m root mean square error (RMSE) | 10–30 m horizontal resolution 1–10 cm vertical resolution | once | Yearly |
| | NDVI layer | 20–30–60 m | 5–10 m | Monthly | Every 2 weeks in late winter/early summer, every 3 months the rest of the year |
| | Vegetation and vegetation change layers, including infesting vegetation | 10 m | 3 m | Yearly | Every 3 months |
| | Ground motion layer and data analysis | 5 × 20 m (Not released yet—expected release in 2022) | 10 m | Yearly (Not released yet) | Every 4–6 months |
| | Coastal erosion layer—sedimentary balance and bathymetry | Not released yet | 1–5 m H. res./1 cm V. res. | (Not released yet) | Every 3 months |
| | Hydrological changes and network changes layers | 2.5 m | 10–30 m (higher is desired) | once | Yearly |
| C3S | Hydrological changes and network changes layers | 10 km | 10–30 m (higher is desired) | This indicator is derived from the daily series and represents statistics over a long period. As such, it does not have a temporal resolution | Yearly |
| CAMS | 1 to 5 km spatial resolution for built environment | 7 km | 1–5 km | Daily | Daily |
| CEMS | Hydrological forecast information | 5–10 km | 10–30 m (higher is desired) | Every 12 h | Daily |
| CSS | Pollutant concentrations characterisation (hyperspectral capacity required)—oil spill service | 10 m | 10 m | Every 4 days | Every 4 days |

**Table 5.** Description of requirements that are not satisfied by the current Copernicus products and for which future developments are hoped for within the Copernicus programme.

| Requirement | Spatial Resolution Required | Update Frequency Required |
|---|---|---|
| Vector layer of linear element into and surrounding the site (roads, pipelines, water conducts etc.) | 1–5 m | 1 year |
| Soil erosion and rainfall erosion monitoring | 100 m | 5 Years |

At the end of the consultation phase, 41 specific requirements were identified (Figure 4). After that, for each monitoring domain, the matching of the identified requirements with the Copernicus capacity was addressed, identifying 31 requirements partially or completely satisfiable by the current Copernicus products (see Table 3).

Table 3 presents a synthesis of the available Copernicus products that can already support cultural heritage users, as well as products that could support cultural heritage users in the future, if improved.

Table 4 is derived from Table 3 and shows the gap between the current Copernicus products in detail, and the required product that could satisfy the user requirements by improving the current products.

It is important to highlight that the requirement tied to CSS (pollutant concentrations characterisation) is not satisfied by spectral characterisation of the current Copernicus Mission. Considering that the current Copernicus Candidate Mission includes hyperspectral monitoring capacity [20,24], it is possible that this requirement will be satisfied

in the near future. It is also important to highlight some potential services that would fit by characteristics into the Copernicus operational products that refer to the following requirements (Table 5).

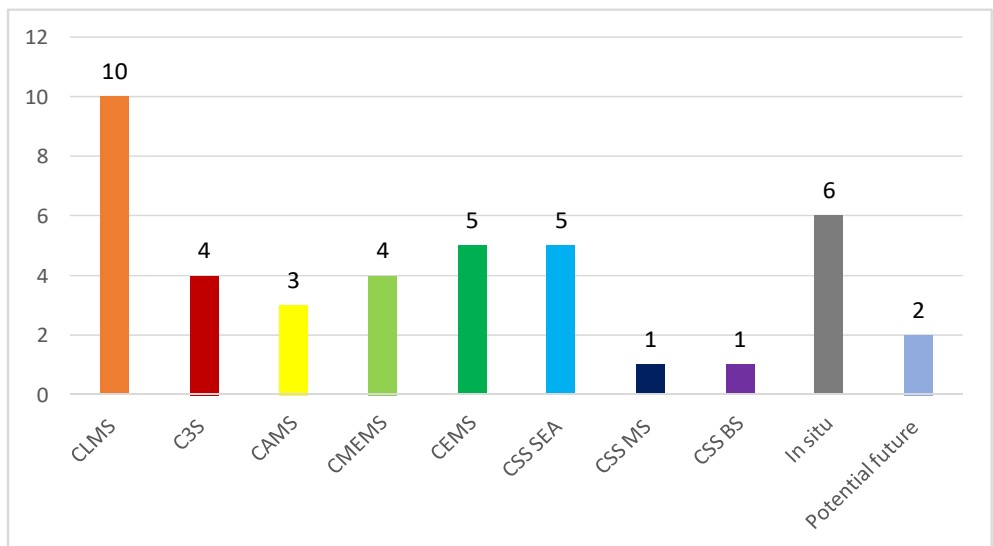

**Figure 4.** Number of requirements per Copernicus service and in situ component.

The analysis shows that most of the routine Copernicus products have the potential to satisfy the requirements. These mostly relate to C3S, CAMS, and CMEMS, and partially relate to CLMS, and refer only to products delivered before 2018 that present spatial resolution (20 m) lower than that currently provided by the sentinel capacity (10 m). Since 2018, the user requirements referring to the Copernicus Land Monitoring Service are mostly satisfied. Additionally, in the land domain, the spatial resolution of existing products could be increased through the use of the Copernicus Contributing Missions (missions from ESA, their member states, EUMETSAT, and other European and international third-party mission operators that make some of their data available for Copernicus). Since CH sites have a limited local coverage, the associated costs for this would be limited, in so far as the data would only be required for specific sites.

With regard to on-demand services, there is a very high matching degree between the identified requirements and most of the Copernicus products delivered. In particular, products have already been released in the fields of cultural heritage monitoring by the Copernicus Security Services—Support to External Actions. Some routine aspects of these services are to some extent hoped to satisfy, even partially, the identified requirements (e.g., building structural movements, velocity, and direction). It is further important to note that the Copernicus Security Service—Border Surveillance, although it does not have any directly associated requirements, also supports vessel detection through existing products in the case of potential cultural heritage smuggling across, particularly in the Mediterranean Sea. The small number of requirements associated with Emergency Management Services reflect a generic need expressed by users (e.g., real-time monitoring of emergency events; pre-event information) that have to be associated, case by case, with different situational crises affecting cultural heritage (e.g., natural and anthropogenic disasters).

Among the requirements expressed by the cultural heritage community, there are some that are not supportable by the Copernicus programme and are not directly related to Earth observation (as listed in Table 6).

**Table 6.** List of unmatched requirements.

| Requirement | Suggested Development Context | Spatial Resolution | Update Frequency | Monitoring Domain |
|---|---|---|---|---|
| Identification of signs of mineralisation | in situ observations | <10 cm | yearly | Observation of changes to the built structure of a CH site |
| Identification of organic change | in situ observations | <10 cm | yearly | Observation of changes to the built structure of a CH site |
| Material composition analysis | VHR imagery and in situ observations | 50 cm | once | Observation of changes to the built structure of a CH site |
| Stratigraphic description of the archaeological site | In situ and geo-gnostic investigations | 1 m | once | Detection of underground archaeological sites through the study of the natural environment |
| Identification of individual layers or stratigraphic units | In situ and geo-gnostic investigations | 1 m | once | Non-destructive analysis of the underground/ underwater positioning of the CH features |
| 3D reconstruction | Support of VHR imagery as ancillary | 50 cm | once | - |
| Metal detecting | S-L-P bands (SAR) in dry soils | 1 m | once per year | Non-destructive analysis of the underground/underwater positioning of the CH features |
| Geology and Petrography | In situ and geo-gnostic investigations | 30 m | once | - |

## 4. Discussion

From the analysis conducted, it is evident that the current Copernicus programme capacities cover a considerable portion of the requirements of the cultural heritage community. Moreover, Copernicus is already playing a crucial role by implementing the paradigm information as a service (IaaS), which allows for the accessibility of complex data in an "easy to read" and "understandable" form. This also supports the target community, which usually has to manage different data sources and related standards that require a broad (horizontal) set of competence and skills, not always possessed by those responsible for the protection and preservation of cultural heritage, especially in the case of implementation of management strategies. Nevertheless, our work highlights that:

1. Some efforts are still required to customise current Copernicus products on the basis of the identified requirements;
2. A unique service access point would be of benefit, to permit users to exploit a single source where Copernicus products and related information are collected and made accessible, with the access to information still being a critical issue;
3. Ready-to-use integrated information layers on land cover/use, geo-hazards, climate, and meteorological conditions, as well as atmospheric parameters, would allow the best understanding of specific phenomena affecting CH sites, and would support users on the basis of a subsidiary model;
4. Access to very high-resolution imagery, to test innovative applications and conduct research for improving monitoring capacities, is required by those representatives of the cultural heritage research community who have the necessary technological skills.

A single infrastructure, such as a thematic Copernicus CH hub, where users can find the required information from the different services, bundled at a unique "access point", would save time searching and retrieving this information. Hub-hosted information should be categorised, should be accessible via standard web services, and should provide a full thematic view of the site or area of interest. Furthermore, in most cases, these approaches would mean that full data downloads (i.e., including all the raw data necessary to synthesise higher-level CH products) would become unnecessary on the end-user side.

With regard to this option, it would also be desirable if it included a cloud computing facility, to support users in deriving information from the combination of available Copernicus products, in a user-friendly environment with a suite of ready-to-use tools.

As an additional consideration, the needs analysis also shows that most of the user requirements fall into the institutional domain, where users are responsible for cultural heritage management and have a responsibility to fulfil national and European obligations and global treaties. In addition, where an institutional requirement is expressed as a continuous public demand, an "anchor tenant" mechanism can foster the development of a more fertile (both public-to-market and subsequently market-to-market) downstream segment.

In order to catalyse the demand and boost the development of downstream cultural heritage services, a market place business model would support the connection of users and providers of geospatial solutions. This would be achieved by using a European institutional demand as an anchor customer, supporting industry and SMEs in responding to institutional operational needs around specific themes [21,25].

Institutional users would benefit from cost-effective information to specifically respond to horizontal needs, such as tourism management, security of sites and visitors, monitoring of wildlife, real-time frequentation statistics for sites, augmented reality, digital 3D model generation, etc.

A Copernicus Market Place model could also play a crucial role for the European assets that are common to all MS, such as:

- Shared processing capabilities, data sets, and models/algorithms (provided not only by Copernicus);
- Common data storage and access (e.g., provided by European DIAS—Data and Information Access Service);
- Unified access monitoring, security, and privacy aspects (according to EU legislation);
- Common services for institutional and commercial users (provided by Copernicus).

This approach could lead to an improved downstream sector with activation of economies of scale (i.e., lower cost per MS for both institutional and private actors), joint R&D benefiting all participating MS, shared IT services and support/maintenance, enhanced international competitiveness, and high-performance computing (HPC) available as a service [21,25].

## 5. Conclusions

The cultural heritage sector stands to benefit greatly from an increased use of remote sensing technologies. Firstly, the latter can potentially provide faster, more economical alternatives to traditional on-the-ground surveying and mapping methods. Moreover, we expect that the use of Copernicus capabilities by cultural heritage stakeholders (including cooperation between the many domains of application represented by the Copernicus user communities) will produce innovative new methods and approaches to cultural heritage management and protection. While there is much technological knowledge already present, there is a lack of operationalisation of methods and tools on an infrastructural level. This is what Copernicus must address in its intended role as an enabling technology programme for the cultural heritage sector. This paper provides information and data with the aim of identifying user requirements and needs, and of producing a detailed analysis of their matching with current Copernicus capabilities. Most importantly, this paper addresses how Copernicus truly caters for the needs of the cultural heritage user community, and presents the potential to enable innovative and value-added (even downstream) products and services. Considering that the majority of the current Copernicus products already satisfy the identified requirements, it appears clear that there is no need for a new core service dedicated to cultural and natural heritage monitoring.

Following these main outcomes, there is high potential for Copernicus to stimulate substantial growth of downstream market services applied to the cultural heritage domain.

The creation of a common platform, where different players (users and providers) can interact for the definition and development of user requirement-based services, would support the market uptake processes.

With regard to this point, from both an economic and social perspective, cultural heritage is already a heavyweight that still offers enormous growth and innovation po-

tential; although, nowadays, it is only partially exploited in both the institutional and commercial frameworks. This importance must be urgently reflected by substantial and, above all, sustainable investment into all of the EU's relevant technological programmes, with Copernicus perhaps being the most obviously important one.

**Author Contributions:** Conceptualization, A.B., N.B., B.D., D.S., A.P.R. and A.T.; methodology, A.B., N.B., B.D., D.S., A.P.R. and A.T.; validation, A.B., N.B., B.D., D.S., A.P.R. and A.T.; formal analysis, A.B., N.B., B.D., D.S., A.P.R. and A.T.; investigation, A.B., N.B., B.D. and D.S.; data curation, A.B., N.B., B.D. and D.S.; writing—original draft preparation, A.B., N.B. and D.S.; writing—review and editing, A.B., N.B., B.D. and D.S.; supervision, A.T. and A.P.R.. All authors have read and agreed to the published version of the manuscript."

**Funding:** This research received no external funding.

**Institutional Review Board Statement:** The study did not require ethical approval.

**Informed Consent Statement:** Not applicable.

**Data Availability Statement:** Repository with row data do not exist; the data used in the manuscript are reported in reference.

**Acknowledgments:** The authors thank the Copernicus Task Force on Cultural Heritage participants, who contributed with national user requirements, and the Task Force chair Salvatore Nastasi, Director General of the Italian Ministry of Culture.

**Conflicts of Interest:** The authors declare no conflict of interest.

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
