# Peer review of "Copernicus in Support of Monitoring, Protection, and Management of Cultural and Natural Heritage"

_sustainability, doi:10.3390/su14052501_

Round 1

Reviewer 1 Report

The proposed paper novelty of the paper is represented by the interaction matrix developed to collect and identify the main gaps regarding data and information monitoring regarding the protection of European Cultural and Natural Heritage. The result of the proposed system manages to identify the user needs from Copernicus monitoring system regarding the daily challenges of preserving and protecting Cultural and Natural Heritage.

The results presented in the research paper are very important to the field of cultural heritage safeguarding, and they are based on the coordination mechanism adopted by the European Copernicus Cultural Heritage Task force. The paper presents synthesized results achieved in term of gap analysis and assess for the current and potential uptake of Copernicus data, services and products aimed to support the process of monitoring and protection of Cultural and Natural Heritage.

The paper is well organized and follows the standard research article structure. The introduction section provides an overview of the constant awareness increasing from the last decades related to the science-based tools used to monitor and protect cultural and natural heritages. The Copernicus program and it`s infrastructure as well as the mechanism adopted by the European Copernicus Cultural Heritage Task force.  The article could include a related work section, as other researchers have identified various aspects regarding the monitoring and protection of CH. The materials and methods section makes use of a flow chart (Figure.1), the Step 6 should be Step 4, as there are two missing steps within the presented flow chart, within page 5 line 129 the hap analysis step is correctly numbered as Step 4). The proposed CCHTF methodology survey was addressed to 48 stakeholders from a total of 14 countries. The results are based on the user need analysis and the requirements definitions. The validation activity and gap analysis have identified 41 specific requirements and for each monitoring domain, the matching of the identified requirements with the Copernicus capacity has been addressed, thus identifying 31 requirements that are either partially or completely satisfiable by the current Copernicus products. The authors have analyzed the gap between the current Copernicus products and the required product that could satisfy the user`s requirements for Copernicus Services and in situ Component. Table 5 highlight the requirements that are not satisfied by the current Copernicus products and Table 6 provides and overview of the user requirement expressed by the CH community that are not supportable by the Copernicus products and not directly related to Earth observation. The discussions and conclusion sections are based on the proposed CCHTF methodology aimed to analyze the user`s needs regarding the monitoring, protection, and management of Cultural and Natural Heritage.

The proposed materials and methods used by the authors are sufficient detailed within the proposed paper. They are based on the EC report analysis and survey conducted by the authors.

The proposed research articles have a small number of references considering the length of the paper, adding a related works section would facilitate a better engagement with sources aimed at monitoring, protection and most importantly the management of Cultural and Natural Heritage assets.  Most of the references are represented by research articles and EC reports that have been published within the last five years.

The overall merit of the proposed paper is represented by the recommendation and the implementation of infrastructural solutions to improve Copernicus Services data and information access by Cultural Heritage users.

A suggestion to improve the proposed is the following:

  • The related works section would increase the engagement with sources as well as more recent scholarship regarding the daily challenges of preserving and protecting Cultural and Natural Heritage. 

The proposed methodological approach has only four steps as it is illustrated in the Flow chart from Figure 1. Figure 1 requires minor correction (Step 4 instead of Step 6 for the Gap analysis) 

Author Response

Dear Reviewer,

thanks for your valuable comments for revision.

We accept all your points. Please see the attachement containing the reviewed file.

Best regards

Nico Bonora

Reviewer 2 Report

Dear Authors,

The article Copernicus in support of monitoring, protection and management of Cultural and Natural Heritage (article ID 1596320) presents insight from the institution's needs in terms of Copernicus products dealing with cultural and natural heritage. The article is well-written and documented. At this stage, I would recommend some Minor revisions. There are some comments and suggestions that authors need to address:

Please, correct the way you wrote your affiliations.

Since you have included “climate change” in the keywords, I think you could also include it somewhere in the abstract

More appropriate references are needed (especially in the Introduction section) in order to refer to cultural and natural heritage. There is more recent literature referring to climate change impacts on cultural heritage. I can suggest (but not limited to):

https://doi.org/10.1002/wcc.710

https://doi.org/10.1073/pnas.1912246117

https://doi.org/10.1007/s10584-017-1929-9

L43-45: please, include at least a reference for each process. There is plenty of literature to do that

L63: please, write from what the acronym stands from

L97: what does PwC mean?

Figure 3: it is really difficult to read the figure. Please, increase the font and/or make the text bold

The author’s contribution is missing

Good luck with the revision!

Kind regards!

Author Response

(The authors gave the same response as above.)

Reviewer 3 Report

This is a very well-done overview and evaluation of Copernicus Services' data for European Cultural and Natural Heritage preservation needs. It offers a profound gap analysis and recommendations for the improvement of Copernicus Services data access for users in the Cultural Heritage field.

Author Response

(The authors gave the same response as above.)
